# Genetic Containment for Molecular Farming

**DOI:** 10.3390/plants11182436

**Published:** 2022-09-19

**Authors:** Amy L. Klocko

**Affiliations:** Department of Biology, University of Colorado Colorado Springs, Colorado Springs, CO 80918, USA; aklocko2@uccs.edu; Tel.: +1-(719)-255-4907

**Keywords:** molecular farming, plant expression systems, gene flow, transgenic plants, biopharmaceuticals, genetic modification

## Abstract

Plant molecular farming can provide humans with a wide variety of plant-based products including vaccines, therapeutics, polymers, industrial enzymes, and more. Some of these products, such as Taxol, are produced by endogenous plant genes, while many others require addition of genes by artificial gene transfer. Thus, some molecular farming plants are transgenic (or cisgenic), while others are not. Both the transgenic nature of many molecular farming plants and the fact that the products generated are of high-value and specific in purpose mean it is essential to prevent accidental cross-over of molecular farming plants and products into food or feed. Such mingling could occur either by gene flow during plant growth and harvest or by human errors in material handling. One simple approach to mitigate possible transfer would be to use only non-food non-feed species for molecular farming purposes. However, given the extent of molecular farming products in development, testing, or approval that do utilize food or feed crops, a ban on use of these species would be challenging to implement. Therefore, other approaches will need to be considered for mitigation of cross-flow between molecular farming and non-molecular-farming plants. This review summarized some of the production systems available for molecular farming purposes and options to implement or improve plant containment.

## 1. Introduction

Molecular farming is the use of plants to produce high-value molecular products. The first published example was production of a human growth hormone fusion protein in *Nicotiana tabacum* (tobacco) and *Helianthus annus* (sunflower) [1]. Since that time, many products have been developed; these include numerous human therapeutics including edible vaccines, human growth hormone, the anticancer drug paclitaxel (commonly referred to as Taxol, this term hereafter), and the malaria treatment artemisinin (see Table 1 for specific examples, as reviewed in [2,3,4,5]). Other applications include edible vaccines for animals, industrial enzymes such as expansin cellulose degradation, polymer production such as spider silk and biodegradable plastics, [6,7,8,9,10]. While some of these products, including Taxol, can be isolated from wild species, integrated production approaches often include transfer of genes into other species. For reviews and extensive examples of products obtained via molecular farming see [11,12,13,14,15,16].

The first commercialized molecular farming product dates to 1997, when avidin, a diagnostic protein, was produced in the seeds of transgenic *Zea mays* (corn) [17]. The first approved molecular farming product for use in animals occurred in 2006, for *Nicotiana tabacum* (tobacco) cell suspension [8] produced poultry vaccine (approved by the USDA but not marketed, reviewed [18]). Approval for the first molecular farming product for human use occurred in 2012 for Protalix BioTherapeutics of Carmiel’s Elelyso, an enzyme produced in *Daucus carota* (carrot) cell suspension used for the treatment of Gaucher’s disease [14,19]. A wide array of other molecular farming products have been developed, many of which have reached clinical trial stages for human use (reviewed in [18,20]). 

**Table 1 plants-11-02436-t001:** Examples of molecular farming expression systems and products.

Type of Expression System	Species	Product	Application(s)	Reference
Whole plant—stable Food crops	*Z. mays* (corn)	expansin (enzyme)	Biofuel production	[10]
	*Z. mays* (corn)	Avidin	Diagnostics	[17]
	*Spinacia oleracea* (spinach)	Rabies glycoprotein	Vaccine	[21]
	*O. sativa* (rice)	Lysozyme	Numerous	[22]
Whole plant—stable Non-food crops	*N. tabacum* (tobacco)	Polyhydroxybuterate	Plastic production	[6]
	*L. minor L.* (duckweed)	Peptide Me2	Avian flu vaccine	[7]
	*N. tabacum*	185-kD antigen I/II (antibody)	Anti-cavity	[23]
	*N. tabacum*	Insulin	Diabetes treatment	[24]
Whole plant—transient	*N. benthamiana*	ZMapp^TM^	Therapeutic	[11]
	*Pisum sativum* (pea)	Human growth hormone	Therapeutic	[25]
	*Lactuca sativa L.* (lettuce)	hOAT (antibodies)	Therapeutic	[26]
	*N. benthamiana*	aprotinin (serine protease inhibitor)	Therapeutic	[27]
Culture of hairy roots	*N. tabacum*	M12 (antibody)	Therapeutic	[28]
	*N. tabacum*	interleukin-2	Therapeutic	[29]
	*Solanum tuberosum* (potato)	hepatitis-B surface antigen	Vaccine	[30]
	*Artemisia*	Artemisinin	Malaria treatment	[3]
Cell suspension	*Daucus carota* (carrot)	Elelyso (Enzyme)	Gaucher’s disease treatment	[19]
	*N. tabacum*	Human growth hormone	Therapeutic	[5]
	*P. patens* (moss)	Factor H	Therapeutic	[31]
	*Taxus ssp.*	Taxol	Cancer treatment	[4]

This table contains a few select examples to highlight the types of systems of expression, species, and products, it is by no means exhaustive.

The lag time between initial product development in the 1980s to first commercial approval in the 1990s for animals and 2010s for humans highlights some of the hurdles of molecular farming. These include the challenge of moving from initial development and proof-of-concept experiments to actual market (reviewed in [32]). In brief, molecular farming underwent an enormous expansion in the 1990s followed by a rapid decline due to many factors including regulatory considerations, social acceptance issues, economic considerations, and lack of focus in product types, followed by a more recent increase in molecular farming products (reviewed in [14,32]).

There are numerous benefits to using plants for production of high-value molecules (for reviews see [11,33,34]). Plants are easy to grow and can have high yields of the desired product (reviewed in [34]). For human health applications, there is little risk of carryover of human pathogens or infectious agents or bacterial toxins. One unique feature is that plant-produced vaccines provide an egg-free alternative production platform, which would avoid issues with vaccine production during chicken egg shortages [11]. Additionally, some childhood vaccines are egg-produced, such influenza, and do carry a small risk for those known to be severely allergic to eggs, so a plant-based alternative could be helpful [35]. Of course, a change to a plant-based production system does not eliminate allergy risk, as food allergies to *Arachis hypogaea* (peanut) *Triticum* (wheat) and *Glycine max* (soy) are common food worldwide, which reduces the suitability of these species for molecular farming [36]. A plant based approach is thought be more affordable in developing countries than microbial or animal cell culture (reviewed in [37]). Molecular farming can also have a very fast implementation for response to infectious disease outbreak, such as the successful use of ZMapp^TM^ against Ebola and the ongoing clinical trials of this product (reviewed in [11]). However, there are significant drawbacks to molecular farming, these include of regulatory compliance, such as lack of regulatory structure to accommodate plant-produced products, issues of post-translational protein glycosylation, and key for this review, containment of molecular farming plants and products (for reviews see [16,34]). The focus of this review is on choice of plant type and plant growth system, and pros and cons of each, with particular attention paid to genetic containment strategies.

## 2. Plant Containment

In broad terms, containment of plants refers to both the physical plant itself and reproductive propagules that can be produced by the plant, such as vegetative fragments capable of regenerating a new plant (ex. roots or tubers), seeds, pollen, and spores. Pollen is of particular concern as pollen can readily disperse to nearby plants, leading to gene flow into neighboring plantings of sexually compatible species, or to wild or feral relatives [38]. As molecular farming plants generally produce high-value products, many of which are developed for use in human health, it is imperative to avoid unintended co-mingling with non-molecular farming plants during production, harvest, transport, and processing. Unfortunately, historical incidents illustrate that human error can easily lead to crossover of molecular farming products into food and feed streams (for an overview with specific examples see [39]). Much of what we currently know about plant gene flow and genetic containment options comes from work on genetically modified (GM) plants. Many plants used in molecular farming are indeed GM, as they had one or more exogenous genes added to allow for production of the desired biomolecule(s). Thus, there is potential for both gene flow containing one or more transgenes, and material flow containing the desired biomolecule(s) from these molecular farming plants.

Plant containment can be implemented via several methods, more than one of which can be utilized at a time (Figure 1). These can include physical, genetic, temporal, developmental strategies and more, for reviews see [38,39,40,41]. Physical approaches include planting the molecular farming crop a set distance away from potentially compatible plants, implementing border rows of standard plants around the molecular farming crop, removing reproductive structures prior to maturity such as detasseling corn, or growing plants in an enclosed environment such as a greenhouse or growth chamber. Genetic specific containment options include using transient transformation such that any transgenes are not passed on to potential offspring, growing sterile cultivars, using plants with high levels of self-pollination and thus low levels of out-crossing, creating sterile hybrids, using plastid transformation (such that potential flow via pollen is greatly reduced), growing dioecious plants to reduce gene flow potential, or engineering sterility [40,42] Other options include expression in vegetative tissues such as leaves, allowing for harvest prior to maturity, or by implementing organ culture or cell culture of the plant material. These last two options utilize growth in a chamber, which adds an additional layer of physical containment. Another possibility is to use phenotypically distinct plants for molecular farming, such that the plant (prior to a certain degree of processing) is readily recognizable [43]. Purple fruited *Solanum lycopersicum* (tomatoes) were proposed for this purpose and used for antibody production [44]. However, there are now both conventional and GE purple tomatoes developed for non-molecular farming purposes, so this trait is no longer suitable as an indicator of molecular farming tomatoes [45]. Some of these approaches will not work well for all molecular farming plants as sometimes the wanted product is the seed, the formation of which generally requires sexual reproduction.

After the plant is produced and grown, containment must extend to harvest, transport, processing and the product reaching consumers, all separate from food and feed uses of plants. In general, there is interest in creating a specific product stream for molecular farming. Such an approach would perhaps improve consumer perception of molecular farming as safe, and lead to increased acceptance of products.

### Systems for Expression and Growth

There are many plant types and growth system options available for molecular farming (Figure 2). Starting material is either transgenic or cisgenic plants (achieved either via stable or transient transformation) or plant material with endogenous genes. Here, the term cisgenic refers to plants which have had one or more genes from breeding compatible species directly transferred to their genome, rather than obtained via cross breeding. This approach has been used to obtain improved disease resistance in *Solanum tuberosum* (potato)_ and *Malus domestica* (apple) and could be used to add traits of interest for molecular farming [46,47]. A challenge of using cisgenic crops for any purpose is the complexity of regulations for such plants, which impacts their marketability, especially for global usage [48].

The choice of plant ranges from edible food and feed crop plants, non-edible (to human) crop plants, to model organisms and even wild species. Once the starting material is selected, then there are choices of systems for growth. These include open field cultivation, enclosed greenhouses or growth chamber, tissue culture of plant organs, or culture of suspended cells. Each of these systems for growth comes with relative levels of potential escape from containment, either via non-propagatable plant material (which may still be bioactive) or via gene flow. In this setting, “escape” refers to the plant itself, or its genes (in the case of transgenic plants), leaving the established site of cultivation and becoming mingled into other plants or plant materials. Open field cultivation represents a high level of potential escape from containment, while cell culture represents a low level of potential escape. During open field cultivation, possible routes of escape include, but are not limited to, pollen flow, seed drop, and accidental co-mingling during cultivation, harvest, transport, or processing. For cell culture, the main possible route of escape would be due to human error during downstream processing, as cell culture materials need highly specialized care.

While any starting material and growth system pairing is possible, it is very typical to generate stable transgenic plants grown as whole plants in a greenhouse or growth room setting for expressing and studying proteins [34]. Once successful expression of the desired molecule is obtained, then other options for production are explored, such as changing from model systems to crops, moving from greenhouse to field, or scaling up the level of production.

The use of food and/or feed plants for molecular farming has both great potential for success and great risk. Food crops are of particular interest for production of edible vaccines for both human and animal use (for numerous examples see [43]). Possible species of interest include *Oryza sativa* (rice), *Solanum lycopersicum* (tomato), *Zea mays* (corn), *Daucus carota* (carrot), potato and more [34,43]. Advantages to these species includes well-developed productions systems as humans commonly cultivate these plants, well-characterized gene expression systems, and for edible vaccines meant to be consumed directly from the plant, no product purification is needed [14]. Indeed, the first edible plant-produced vaccine (it was not commercialized) was expressed in the tubers of potato, however, it is neither typical nor advisable to eat raw potato tuber, and cooking inactivates the vaccine protein [2]. There was interest in transitioning to the related *Solanaceae* species tomato, as tomato fruits and fruit products are commonly consumed raw. As other plants such as carrot, *Lactuca sativa L.* (lettuce), and *Musa acuminata* (banana), are often eaten raw, these represent other options for production of edible vaccines. Some downsides to use of food and feed plants include regulatory barriers to production, shelf life of edible products, uncertainly of dosing of edible vaccines, and consumer willingness to consume a GM plant [4,43,49]. However, it is likely that some individuals, particularly children, may be far more willing to eat a snack than submit to a needle poke.

Many of the food and feed crops of interest for molecular farming are grains, including wheat, corn, *Hordeum vulgare* (barley), and *Oryza sativa* (rice) [50]. Here, the seed endosperm and its ability to store large amounts of proteins and other compounds is used to obtain a high yield. Here, endosperm-specific promoters are used to drive gene expression (for specific promoter examples, see [50]). Rice is of particular interest for this process as rice plants tend to be self-pollinating, has a high yield, and has well-developed transformation systems [51,52]. Dry seeds are also very stable, which allows for storage and future processing, and eliminates the need for immediate usage, processing, or drying of vegetative materials [53]. However, in most cases these seeds would be no different in appearance from seeds intended for food or feed, and human error could lead to co-mingling of molecular farming seeds with non-molecular farming seeds.

A main concern for use of food and feed plants for production of molecular farming products is containment, both in terms of accidental crossover into food or feed streams, and the potential for gene flow from molecular farming plants to non- molecular-farming plants (reviewed in [43]). Typically, non-molecular farming food and feed plants are grown in open fields, and this is a standard approach for large-scale plant cultivation. However, this approach, without added containment, represents a high risk of potential escape. There is precedence for concern of field-grown plants as a prior example of molecular farming corn production led to accidental crossover into a food/feed crop and potential gene flow into a nearby corn planting (reviewed in [39,54,55]). To summarize, in 2002, growers of ProdiGene corn engineered to express a *Sus scrofa domesticus* (domestic pig) vaccine failed to follow guidelines for containment. Corn seeds present in the field from the previous growing season germinated as “volunteers” in a crop of soybean when no crop should have been present as the field was meant to be fallow (unplanted) and surveyed for just such corn seedlings, which were to be removed and destroyed. Instead, presence of the sprouted corn meant that 12,000 tons of soybean plants needed collection and destruction. In a second location, the ProdiGene corn was grown near another field of corn, which may have led to gene flow from the molecular farming corn to the non-molecular-farming corn via wind-born pollen. These potential pollen-receiving corn plants were also destroyed. In both cases, human error in not following existing guidelines led to actual or potential gene flow. To mitigate this issue, it has been proposed that if molecular farming is done in food crops these plants ought to be grown away from centers of production for each particular crop [56]. It is proposed than even reduced yield obtained from growing such plants outside of their ideal locations should still lead to enough product for use [56]. Other options are to change how the plant is grown, such as organ culture or cell suspension, leading to a high level of physical and genetic containment (see below for more details).

Another possible route for obtaining high levels of expression of the desired molecular farming product with increased levels of containment is transient expression (reviewed in [32,57]). Here, the plant is grown in physical containment (such as a greenhouse), then modified with a viral or bacterial vector containing the gene(s) of interest. In this situation, the somatic plant cells are modified to express the protein, but the germline is unchanged. Thus, the risk of gene flow via sexual propagules is reduced. Asexual spread is still possible but is limited by the physical enclosure of the plants to an indoor environment. In such situations, temporal containment can also be utilized as the plant can be grown during times of the year when outdoor cultivation may be limited by weather, further decreasing the possibility of spread to the environment or gene flow to compatible plants.

One advance to transient transformation is speed. The yield can be obtained a very short time frame post-inoculation (days for *Nicotiana benthamiana*, a close relative of tobacco, and for tobacco varieties), making this system much faster than creating stable transgenics [57]. It is estimated that plant-based transient expression could decrease the time needed to produce a vaccine or other therapeutics from months to weeks, making it of particular interest as a quick response to disease outbreaks [14]. There is evidence that such an approach can be successful. Part of the response to the 2014–2016 outbreak of Ebola included production and successful use of ZMapp^TM^, a combination of therapeutic antibodies produced via transient expression in *N. benthamiana*, which led to survival of the small group of patients tested [11]. The hope is that the relevant plant material could be grown and the expressed vaccine components immediately used at the site of the outbreak, rather than relying on transport of pre-made vaccines, many of which require refrigeration [11]. If there is flexibility in the age at which plants could be transiently transformed, then plants could be grown and kept in reserve to be used as needed, rather than needing to plant seeds and wait, which could lead to quick production as needed. The recent global COVID-19 pandemic has only served to illustrate some of the difficulties of vaccine production and distribution logistics, a rapid local production system should help address some of these concerns [58,59]. The development of vaccines against COVID-19 includes notable examples produced with a plant-based transient expression system. A tobacco-produced fusion protein has undergone phase 2 clinical trials (NCT04473690), while a full-length protein expressed in *N. benthamiana* completed phase 3 clinical trials (NCT04636697) and was approved for use in humans by the Canadian government [60]. These successes show that plant produced vaccines from transient expression are feasible, effective, and appliable. The downsides are that transient expression does not work well for all species, large-scale production requires an investment of time and effort, and results in a single harvest of material.

There is great interest in using non-edible (to humans) crop plants for molecular farming purposes. Tobacco and its relatives are particularly amenable to molecular farming due to its rapid growth, ease of stable and transient transformation, high product yield, and ability to be grown as whole plants, hairy roots, and cell suspension [57]. As a non-food and non-feed plant, tobacco is unlikely to accidentally enter human food production. Drawbacks to tobacco include a negative connotation associated with nicotine usage, which does complicate regulatory issues of edible components derived from tobacco [57]. Tobacco is widely grown for human use so gene flow in an open-field situation is possible. A low-alkaloid male-sterile hybrid variety where the product accumulates in leaf tissues has been developed to address both these drawbacks [61] Additionally, the tractability of tobacco to transient expression, or to growth in cell culture, and the accumulation of the desired product in vegetative structures (meaning harvest can occur prior to flowering and seed production) all reduce likelihood of escape from containment.

Another option for obtaining plant produced materials is to use model species, such as *Arabidopsis thaliana* (*Arabidopsis*), *Lemna* (duckweed), and *Physcomitrella patens* (moss) for molecular farming (for examples see [31,34,62,63]). The use of model plants for molecular farming has many benefits. Such plants are highly unlikely to accidentally enter the food stream, due to these not being food species and that they are generally grown in physically enclosed spaces such as greenhouses and growth chambers, rather than in an open field. Moss and *Lemna* can be easily multiplied vegetatively, and thus are less likely to have unintentional gene flow via free spores (moss) or pollen (*Lemna* and *Arabidopsis*) than species mainly propagated sexually. These species also have well-characterized genetics and transformation systems. Drawbacks to using model systems include that, like all plants, these species have limitations. One example was an attempt to produce Taxol in *Arabidopsis* by adding the relevant biosynthesis pathway genes. While Taxol was successfully produced, both Taxol yield and plant growth were poor, which was thought to be due to Taxol biosynthesis requiring an intermediate molecule, which is also key for other plant compounds [62].

One promising method for obtaining plant produced compounds in a highly contained environment is cell culture, either of hairy roots, or cell suspension. Such an approach could be applied to crop plants, model plants, or even to cultivated non-domestic species. Here, the plant material is grown in sterile conditions with solid or liquid medium providing nutritional support. A high level of physical and genetic containment is achieved by hairy-root culture. Specific efforts underway include hairy root culture of *Artemisia annua* (*Artemisia*), a plant which produces the highly valuable compound artemisinin (AN), which is used to treat Malaria (reviewed in [64]). Whole *Artemisia* plants produce the AN compound in trichomes, small hair-like structures found on leaves. Current growth and production of AN cannot keep pace with Malaria outbreaks and patient needs, so molecular farming is being explored as an option to meet this demand. Initial efforts utilizing transgenic tobacco engineered to biosynthesize AN led to a precursor compound in the biosynthetic pathway [65]. Use of hairy roots of *Artemisia* itself gave good results, thought to be due to AN production in root hairs, which appear to share the same AN biosynthesis pathway as trichomes [64]. It is hoped that AN produced by hairy root culture of *Artemesia* species will be commercialized soon.

Plant cell culture suspension is a similar approach to hairy roots, and one that has already reached commercialization for some products. The first molecular farming product approved for use in humans was produced in cell-suspension of transformed carrot [19]. Another cell culture approach currently underway aims to produce the anti-cancer compound Taxol (reviewed in [4]). Taxol is a high-value compound, used for treatment of many human cancers (reviewed in [4]). The initial source for Taxol is several species of the genus Taxus, commonly referred to as yew trees. Current demand far outstrips production capability of wild yew, as it is estimated that a single patient would require a total amount of Taxol equivalent to that produced by the bark of 8 trees each of 100 years of age [4]. Many alternative productions systems are now in place, including harvest from managed forests, and collection of leaves and twigs (rather than bark) such that the harvest does not kill the trees. This approach is still limited by availability of biomass and seasonal availability of new growth. Currently, successful cell culture of yew is underway commercially [4].

Growing plant organs or cell suspension has many potential benefits. These include creation of a stable supply not dependent on seasonal growth or availability of the needed plant material [66]. This production system also reduces pressures on wild sources for wild plants, a key factor for Taxol production. Production of cell culture an make it easier to meet regulatory requirements (as cells are grown aseptically) rather than in soil or even in rock wool, etc. (reviewed in [18]). Typically, no pesticides or herbicides are required in aseptic plant culture, so these compounds are not carried over into the final product. From a containment point of view, cell culture systems are an ideal platform for production. As the plant material is either cells dispersed in culture medium or organs in culture vessels, the plants are extremely isolated from all other plants, highly unlikely to survive and propagate on their own, or to accidentally enter the food or feed product streams. No sexual reproductive structures are produced, decreasing the potential for gene flow. In short, it would take a massive amount of human error for these materials to escape containment. Some downsides to cell culture are the expert knowledge, expense, and infrastructure needed to implement such production systems, and the optimization required to obtain a useable yield.

There are many options available for mitigating potential flow from molecular farming plants to other plants. These include genetic methods, type of plant chosen, physical containment, developmental containment, and temporal (timing). More than one approach can be used simultaneously to obtain multiple levels of containment.

Many molecular farming products rely upon the addition and expression of transgenes, via either stable or transient plant transformation approaches. As a note, transient transformation allows for production in the current generation of plants but is not inherited, which lowers the risk of release. Other molecular farming products make use of endogenous genes. Once the plants are obtained, they can be grown outside in field conditions, inside a greenhouse or growth chamber, as a plant organ in tissue culture (e.g., hairy root cultivation), or as a cell suspension in tissue culture. Each of these growth options comes with a relative level of risk of release from containment.

## 3. Conclusions

Molecular farming has great potential to expand the options available for production of valuable products. The variety of options for plant species, methods of expression, and systems of growth adds to the potential utility of this approach, but also is adds to the complexity. Current and previous production choices illustrate some of the breadth previously achieved. For example, the planting of stably transformed transgenic wind-pollinated corn in an open field near other corn plantings represented a high level of risk of escape from containment leading to actual escape (as illustrated by the ProdiGene incident), while stably transformed transgenic cell culture of carrot suspension cells is illustrative of a much lower level of risk and a product currently on market. In both cases, a food crop was used for production, a key difference was how the plants were grown, leading to a different degree of containment and potential for escape. Overall, there is a need for oversight and planning along all steps of the process, along with consideration of regulations, practicalities and market potential. It is important to have good alignment between the species chosen, mode of expression, system for plant growth, and chosen containment strategy. The myriad of molecular farming products on the market, undergoing trials, in early stages of development, tested as proof-of-concept, or scrapped along with way shows that molecular farming is likely to be an important part of the biotechnology landscape.

## Figures and Tables

**Figure 1 plants-11-02436-f001:**
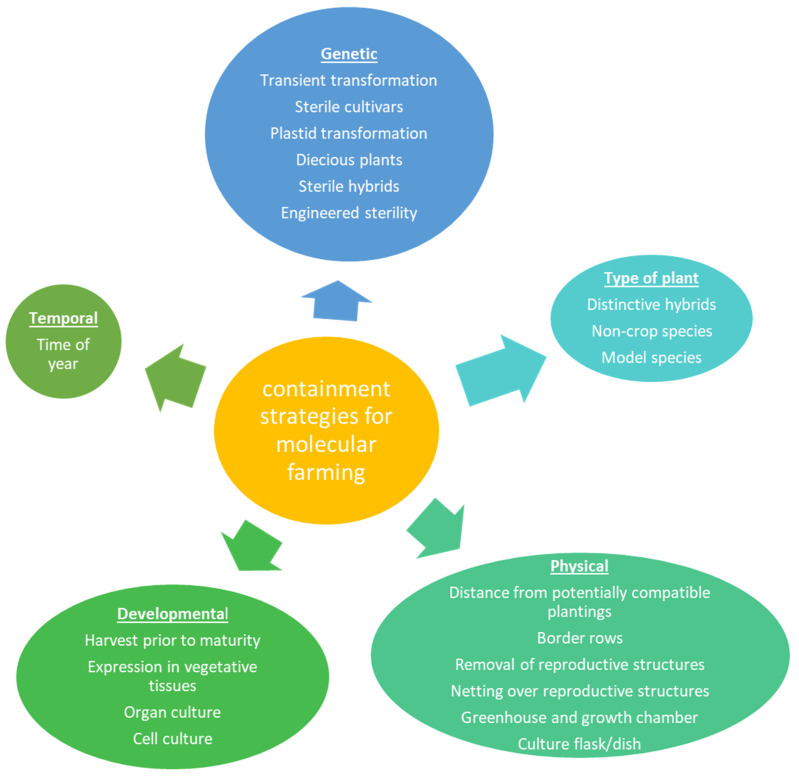
Containment strategies for molecular farming.

**Figure 2 plants-11-02436-f002:**
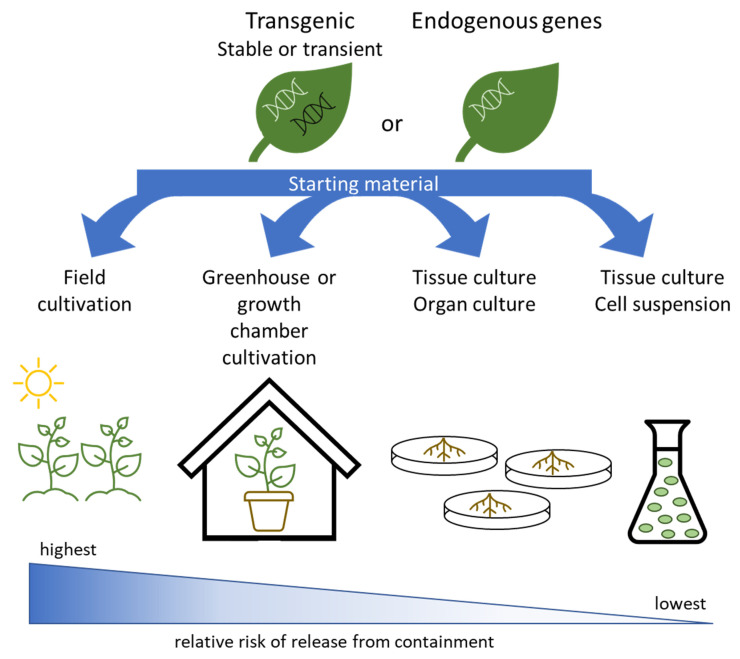
Options for production of molecular farming products.

## Data Availability

Not applicable.

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
