# Peer review of "Genetic Containment for Molecular Farming"

_plants, 2022, doi:10.3390/plants11182436_

Round 1

Reviewer 1 Report

The review entitled “Genetic Containment for Molecular Farming” relates a revision of plants used for molecular farming, and how different developments have been produced through molecular techniques that have reached commercial stages or pre-scaling. The work is appropriate but need some improvements.

Specific improvements:

Lane 9: avoid the use of word “created”, is a controversial term in science, other alternatives are “developed” or “generated”

Lane 10: explain better the inclusion of foreigner genes, if you read the sentence, it seems that a simple mixture is made and that does not reflect the complexity of the genetic transformation, regardless of the technique to be used.

Lane 30: same than lane 9

Lane 30: please develop better the term “reagents”, for experimental conditions “reagent” is a general term, include a lot of different organic and inorganic compounds.

Lane 63: Expand better the idea of risks, plants have allergens for example.

Lane 122: include examples of “cis”-genic GMOs

General improvements:

Please add a table with concrete examples of molecular farming and include a paragraph of other techniques to modulate gene expression relative to molecules of interest (elicitors, primming, or others).

Reviewer 2 Report

The current manuscript reviewed pros and cons of molecular farming as a rapidly developing field for the production of high-value recombinant proteins or metabolites, as well as challenges it's facing. The author mainly focused on the strategies and features of molecular farming-based production from a biosafety point of view, with special emphasis placed on biomaterial containment. Indeed, there is still a long way to the commercialization of most molecular farming-based products and their acceptance by most consumers due to the lack of standardized industrial production pipelines or safety standards for various systems. Nevertheless, with the great progress already made towards the development of molecular farming we believe all the hurdles will be overcome in the years to come.

I do have several minor questions and concerns. My apologies if I was misunderstanding or mistaken.  

1. I might take issue with author's perspective that the plant-produced egg-free vaccine alternative is considered as "nuance". Instead, I think this is a unique feature of molecular farming that is superior to other expression systems. 

2. Some sporadic typos, such as line 139-140.

3. For the COVID19 part, I think there could be some more to put in here to emphasize the advantage of transient expression system, such as the plant-produced ACE2-Fc fusion protein as a potential therapeutic candidate against SARS-CoV-2. A tobacco-produced vaccine candidate based on the SARS-CoV-2 protein S1 subunit was developed by Kentucky Bioprocessing and has undergone Phase 2 clinical trials (NCT04473690). A COVID-19 VLP vaccine presenting the full-length S glycoprotein produced in N. benthamiana by Medicago finished its Phase 3 trials (NCT04636697) and was approved by Health Canada in February 2020 (under the name Covifenz) as the first plant-based vaccine to be approved for use in humans.

4. For the cell culture part, I think the Protalix's Elelyso can also be used as a milestone example since this is also the first FDA-approved plant-made therapeutic protein to treat human disease.

5. I feel some information in table 1 could be more specific. For example, the Enzyme produced in Daucus carota (carrot) could be specified as beta-glucocerebrosidase (marketed by Protalix Biotherapeutics under the name of Elelyso). The same applies to "antigen" or "antibody".

Reviewer 3 Report

Type: Review

Title: Genetic Containment for Molecular Farming

The present manuscript summarizes the production systems available for molecular agriculture and the options for implementing or enhancing plant containment.  It is a very interesting topic, and the author combined/presented relevant information on the subject.

The writing needs some work, and many paragraphs don't have a good connection between them, making a non-flow reading. The main concern about this manuscript is that reviews should present mostly primary research as references, and I could count 25 out of 57 references are other review papers. I see this as “taking away” the other review authors, from the previous publications, a their future citations. The author needs to present more primary research papers. Additionally, only 11 references out of the 57 are from 2018-up and all of them are again review papers. A review paper should present much more recent data and updates on the subject.

Abstract

Text seems to be “broken” with very short sentences that switches subjects all of the sudden.

Line 11:13: This sentence needs to be re-written “Both the transgenic nature of many molecular farming plants and the fact that the products produced are of high-value and specific in purpose means it essential to prevent accidental crossover of plants or products into food or feed production.”

Line 16:18: This sentence needs to be re-written “However, given the large number of molecular farming products in development, testing, or approval that do utilize food or feed species, such a ban would be challenging to implement.”

Line 8: extra space between “.” and “Some”

Actually, it seems that there is an extra space at the beginning of each new sentence in the manuscript.

Line 19: Erase “will” and use “summarized”

Line 23: The word “containment” is already present in the title. Keywords are supposed to be different from the ones used in the title to increase search returns and so give your paper more visibility. Erase “containment” from keywords or replace it.

1.      Introduction

Line 27: Correct “product” to “produce”

Line 33: “[2-5]” references 3 and 4 are already reviewed. It is like “stealing” these authors' future citations.

Line 47: close parenthesis.

Line 69: You have “include” and “including” in the same sentence.

2.       plant containment

Line 74: correct title “p” to uppercase.

Line 107-110: This sentence needs to be re-written “Purple fruited Solanum lycopersicum (tomatoes) were proposed for this purpose, however, there are now both conventional and GE purple tomatoes developed for non-molecular farming purposes, so such fruits are no longer as unique as they were previously [32, 33].”

2.1. systems for expression and growth

Line 119: correct title “s” to uppercase.

Line 127: You mentioned “comes with relative levels of potential escape”. What means high and low levels? You need to be more specific here. Explain what is an escape and give more details about all possible ways of escape.

Line 133: “Once proof of concept is obtained, then other options are 133 explored.” Be clearer here. Which concept? And What are the other options?

Line 145: Could you present other raw consumed veggie products? Carrots and other fruits, for example…

Lines 152 – 177: This paragraph should also mention something about plants with high levels of self-pollination (autogamy) could be a solution for the gene flow issue.

Line 178: Also explain the downsize of using a transient expression, not only that is a single harvest of material but also requires time and effort for the transformation, like hairy root, and in this case a large-scale method.

Line 2253: Transfer this whole part to line 229. “Taxol is a high-value compound, used for treatment of many human cancers (reviewed in [4]). The initial source for Taxol is several species of the genus Taxus, commonly referred to as yew trees. Current demand far outstrips production capability of wild yew, as it is estimated that a single patient would require a total amount of Taxol equivalent to that produced by the bark of 8 trees each of 100 years of age [4]. Many alternative productions systems are now in place, including harvest from managed forests, and collection of leaves and twigs (rather than bark) such that the harvest does not kill the trees. This approach is still limited by availability of biomass and seasonal availability of new growth.”

Line 268: No pesticide or herbicide is required depending on the growth environment and plant/crop used.

Figure 1: Replace "plant of interest" in the figure for "containment strategies for molecular farming" and write a proper legend for the figure that explains it.

Round 2

Reviewer 3 Report

The author has made the requested modifications.